*Review Article*

EMBO
Molecular Medicine

# Exposomics: a review of methodologies, applications, and future directions in molecular medicine

Melissa Wan [iD][1], Elisabeth M Simonin [iD][2], Mary Margaret Johnson[2], Xinyue Zhang [iD][3], Xiangping Lin [iD][4], Peng Gao[5], Chirag J Patel[6], Aroub Yousuf [iD][6], Michael P Snyder [iD][4], Xiumei Hong[7], Xiaobin Wang[7], Vanitha Sampath[2] & Kari C Nadeau [iD][2]✉

## Abstract

**The exposome is the measure of all the exposures of an individual in a lifetime and how those exposures relate to health. Exposomics is the emerging field of research to measure and study the totality of the exposome. Exposomics can assist with molecular medicine by furthering our understanding of how the exposome influences cellular and molecular processes such as gene expression, epigenetic modifications, metabolic pathways, and immune responses. These molecular alterations can aid as biomarkers for the diagnosis, disease prediction, early detection, and treatment and offering new avenues for personalized medicine. Advances in high throughput omics and other technologies as well as increased computational analytics is enabling comprehensive measurement and sophisticated analysis of the exposome to elucidate their cumulative and combined impacts on health, which can enable individuals, communities, and policymakers to create programs, policies, and protections that promote healthier environments and people. This review provides an overview of the potential role of exposomics in molecular medicine, covering its history, methodologies, current research and applications, and future directions.**

**Keywords** Environmental Exposures; Exposomics; Omics; Risk Assessment
**Subject Category** Evolution & Ecology

## Introduction

Human health is affected by both environmental and genetic factors. In the last few decades, transformative advances have been made in measuring and understanding the impacts of genomic variations on health and translating this knowledge to treat diseases. The vast amounts of information in the genome were initially daunting, but advances in technology enabled the sequencing of the 3 billion base pairs in a human genome. In 2003, sequencing the entire human genome took 13 years to complete at a cost of 2.7 billion. Currently, with next-generation sequencing (NGS) technology, it has lowered the cost while increasing the speed of DNA sequencing. The entire genome can now be sequenced in less than a day at a cost of just a few hundred dollars (National Human Genome Research Institute). Genetic advances have enabled the identification of genes associated with diseases such as breast cancer, Huntington's disease, and cystic fibrosis. It has paved the way for early diagnosis and novel and personalized treatments. However, despite the increases in our understanding of genetics and advances in methodologies, our understanding of disease variability is limited leading to increased interest in understanding the environmental effects on disease.

In 2005, Christopher Wild introduced the term "exposome" as an environmental complement to the genome. He defined the exposome as encompassing "… life-course environmental exposures (including lifestyle factors), from the prenatal period onwards" (Wild, 2005). The environment encompasses many types of exposures, including physical (i.e., light, noise, radiation, etc.), biological (i.e., animals, bacteria, fungi, viruses, etc.), chemical exposures (i.e., pesticides, carcinogens, heavy metals, etc.), and socioeconomic, lifestyle, and psychological exposures (Miller, 2014; Wei et al, 2022; Wild, 2012). Miller and Jones further expanded on the concept of the exposome in 2013 as "The cumulative measure of environmental influences and associated biological responses throughout the lifespan, including exposures from the environment, diet, behavior, and endogenous processes" (Miller and Jones, 2014). This definition adds to the burden of measuring "cumulative exposures" and determining "associated biological responses."

Traditional environmental health studies include hypothesis-driven methods which have focused on one or a class of environmental exposures at a few time points. These fail to account for the complex interactions of exposures across the lifespan on human health. Exposomics aims to bridge this gap. Exposomics include the use of high-throughput methodologies and are generally agnostic and data-driven rather than hypothesis-driven. It integrates data from different methodologies and time points to comprehensively and cumulatively determine the impact of the

[1]Harvard Chan Occupational and Environmental Medicine, Harvard T. H. Chan School of Public Health, Boston, MA 02115, USA. [2]Department of Environmental Health, Harvard T. H. Chan School of Public Health, Boston, MA 02115, USA. [3]Cardiovascular Institute Operations, Stanford University, Palo Alto, CA 94305, USA. [4]Department of Genetics, Stanford University, Stanford, CA 94305, USA. [5]School of Public Health, University of Pittsburg, Pittsburgh, PA 15261, USA. [6]Harvard Medical School, Boston, MA 02115, USA. [7]Center on Early Life Origins of Disease, Department of Population, Family and Reproductive Health, John Hopkins Bloomberg School of Public Health, Baltimore, MD, USA. ✉E-mail: knadeau@hsph.harvard.edu

| Glossary | | | |
|---|---|---|---|
| Exposome | The measure of all the exposures of an individual in a lifetime and how those exposures relate to health | Machine learning | Machine learning uses technologies and algorithms that enable systems to learn through pattern recognition and make decisions without being programmed. |
| Exposomics | The emerging field of research to measure and study the totality of the exposome. | Deep learning | It is an artificial intelligence (AI) method that uses artificial neural networks to process data in a way inspired by the human brain. |
| Omics technologies | "omics" technologies aim to studying the totality of specific factors (e.g., genes, mRNA, protein) within a cell, tissue or organism. | | |

exposome on health, assess risk, and estimate the burden of environmental disease. Further, it aims to provide insights into the cellular and molecular changes mediated by the exposome. These cellular and molecular alterations can act as biomarkers for diagnosis and offer new avenues for treatment (Steckling et al, 2018). The goal of exposomics is to discover key biomarkers of exposure that enable follow-up hypotheses to be explored regarding sources of exposure, dose–response relationships, mechanisms of action, disease causality and public health interventions. While it is not realistic to map the entirety of the exposome of an individual or a population, the large datasets that are being generated and analyzed are poised to increase our understanding and decrease the health burden associated with environmental exposures.

There have been a number of conferences, programs, and institutes focusing on this important and growing field. In Europe, institutions such as Helmholtz Munich (Helmholtz Munich), multidisciplinary projects such as the Enhanced exposure assessment and omic profiling for high-priority environmental exposures in Europe (EXPOsOMICS) (Vineis et al, 2017), European Human Exposome Network (EHEN) and the International Human Exposome Network are working towards understanding the role of the exposome on health and findings ways to prevent adverse health effects due to environmental pollutants. Horizon Europe is the EU's key funding program for research and innovation and funded the European Human Exposome Network (EHEN) (European Commission; Fayet et al, 2024). A recent European Exposome Symposium in 2023 brought together leading US- and European-based researchers and trainees in the field of exposomics and environmental health to share their latest findings. In the USA, the HERCULES exposome research center (Hercules exposome research center, 2024), the Institute for Exposomic Research at Mount Sinai (Ichan School of Medicine at Mount Sinai, 2024), and the Network for Exposomics in the U.S. (NEXUS) are major exposomic projects (Environmental Factor). However, there are also many other environmental institutions and centers, including the Environmental Protection Agency (EPA) that are also working on exposomics projects.

Exposomics is being fueled by further major technological advances in genetic and other omics technologies and big data analytics (Vermeulen et al, 2020; Vineis et al, 2020). The exposome leaves footprints on these various "omics" layers and plays an essential role in disease etiology (Wu et al, 2023). For example, unlike the static genome, the epigenome is modifiable and can potentially be used as a biomarker or even intervention target for certain exposures. Integrating exposomics in molecular medicine, therefore, offers a new avenue to interpret the disease etiology with the potential to provide novel disease treatment and prevention strategies.

In addition, exposomics is also benefiting from advances in geospatial monitoring, real-time data from personal wearable

devices, and artificial intelligence. Large collaborative and research centers that serve as intellectual hubs or information exchange clearinghouses facilitating sharing of exposomic tools and data are playing an essential role in supporting the progress of exposomic research. These technologies are paving the way for the construction of models of complex biological systems and diseases, often termed "Systems biology (Soerensen et al, 2024)." Exposomics is still at its infancy, and to date, has primarily assisted with finding associations between environmental exposures and biological response, which can assist with hypothesis-driven traditional methods. However, the potential applications using exposomics for human health are numerous and may include diagnosis, prognosis, assessing disease risk, personalized medicine, health disparities, and population health.

In this review, we provide an overview of exposomics and the evolution of the field. We also discuss the techniques and tools used in exposomics research, collaborations in the field, and the promises and challenges of the field.

# Exposomics research

## Methodological approaches: divide and conquer

Christopher Wild further expounded on the concept of the exposome and argued that with the advances in technology, the time was ripe to move the concept to practice. Wild divided the exposome into three broad and overlapping domains of exposure (1) general external environment (e.g., socioeconomic status, education level, environment, climate), (2) specific external environment (e.g., pollutants, nutrition, exercise, smoke exposure, viruses, and bacteria), and (3) internal environment (e.g., physiology, microbiome, age) (Wild, 2012) (Fig. 1). This division of the exposome into different domains provides researchers a manageable means of dividing the enormity of the different exposures. A more complete exposomics picture could then be obtained by integrating the data from the different domains.

Rappaport and Smith, in 2010, proposed another way to approach exposomics research. They contrasted two approaches to characterizing the total exposome, the top-down and the bottom-up approaches (Rappaport and Smith, 2010). The top-down approach measures important exposure-related biomarkers within biospecimens that are biologically relevant and mediate health effects. It does not capture direct measures of exposure and can generate hypotheses regarding exposure and biological responses. It uses untargeted omics methods to measure both exogenous and endogenous exposures. The bottom-up approach comprehensively measures environmental exposures including chemical, physical, and social factors. This data can be obtained

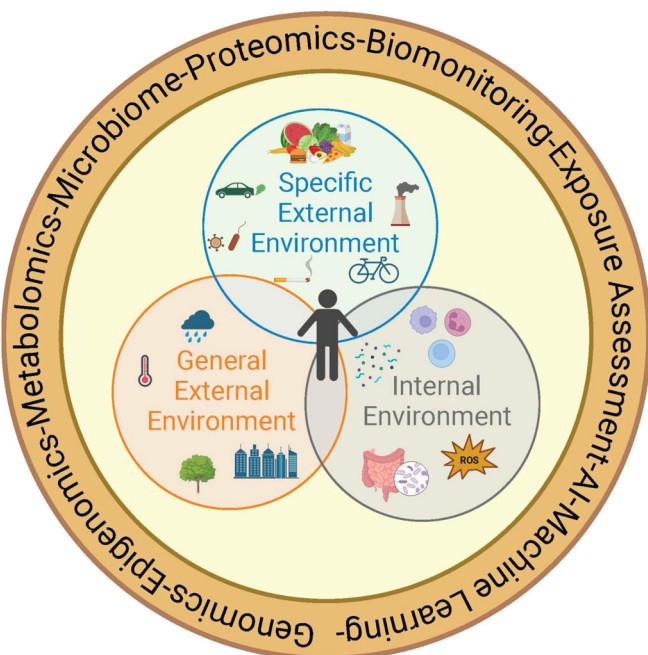

**Figure 1. Molecular medicine and the exposomics.**

The exposome describes the collective exposures from an individual's general external environment (climate, urban, rural), the specific external environment (nutrition, pollutants, exercise, pathogens), and the internal environment, (immune response, oxidative stress, metabolites, microbiome). The exposomics studies the exposome and its impacts on health outcomes through environmental exposure measurements and molecular biological responses through (bio)monitoring and omics technologies, respectively.

through various sources such as geospatial monitoring, surveys, census and other public databases, wearable and portable devices, and social media platforms. While this provides valuable environmental data, it fails to address the internal chemical environment. Both the bottom-up and top-down are complementary, and each approach provides valuable data (Coughlin, 2014; Rappaport and Smith, 2010; Zhang et al, 2021).

## Technologies and tools

Exposomics involve environmental exposure measurements and associated measures of biological responses. It relies on a large number of tools and methodologies such as omics, sensors, geospatial devices, mobile devices, statistical and bioinformatic tools.

### Omics technologies

The rise of "omics" technologies such as genomics, metabolomics, DNA adductomics, proteomics, lipidomics, transcriptomics, microbiomics, and epigenomics offer comprehensive and detailed insights into the genetic, epigenetic, molecular and cellular responses to environmental exposures. These technologies have been made possible by the availability and advancement of high-throughput analytical instruments, such as DNA, RNA, and protein microarrays, nuclear magnetic resonance (NMR) spectroscopy, next-generation sequencing (NGS), bisulfite sequencing, ATAC-seq, single-cell RNA sequencing, chromatin immunoprecipitation

sequencing (ChIP-Seq), and mass spectrometry (MS). Moreover, when combined with additional separation technologies like gas chromatography-mass spectrometry (GC-MS) and liquid chromatography-mass spectrometry (LC-MS), the utility of MS is further enhanced, enabling the detection, quantification, and identification of a broad spectrum of environmental exposures, and the associated biological responses, with increased sensitivity (Dai and Shen, 2022; Karahalil, 2016; Möller et al, 2024; Veenstra, 2021).

There are numerous examples of studies that have associated environmental pollutants with molecular, cellular, genetic, or epigenetic changes. For example, epigenetic alterations have been found in immune cells exposed to ambient air pollutants (Aguilera et al, 2022). A study of Australian women found that long-term exposure to wildfire-related $PM_{2.5}$ was associated with various blood DNA methylation signatures, which were distinct from those associated with non-wildfire-related $PM_{2.5}$ (Xu et al, 2023). An in vitro study using ATAC-seq and RNA-seq found certain genes up-regulated in the ferroptosis signaling pathway in PM2.5-induced asthma exacerbations, which may serve as biomarkers for diagnosis or for targeted therapy (Zhang et al, 2023).

High-resolution metabolomics (HRM) can monitor >1000 small molecules using a relatively small amount of biospecimens and at a comparable cost to traditional methods (Johnson et al, 2010; Jones, 2016). It offers unparalleled qualitative and quantitative analyses of small molecules in biofluids and provides insights into individual metabolic responses associated with different environmental exposures (González-Domínguez et al, 2020; Soltow et al, 2013; Walker et al, 2019). A metabolome-wide association study found that exposure to per- and polyfluoroalkyl substances was associated with alterations in amino acid metabolism and lipid metabolism in adolescents and young adults (Goodrich et al, 2023). A recent prospective birth cohort study found that prenatal exposure to PFAS was associated with cord blood metabolomic perturbations (Li et al, 2024b). In another study, urine samples were collected from sugarcane workers to determine environmental exposures leading to the high rate of kidney disease seen in these workers. Samples underwent exposomic and untargeted metabolomic analysis. The study results suggest that silica and certain pesticides were significantly elevated in the urine of sugarcane workers. These exposures may provide insight into early warning signs of kidney injury and may help explain the increased incidence of chronic kidney disease among these workers (Stem et al, 2024). Other metabolomic studies have identified metabolites prognostic of increased risk of diseases such as diabetes and prostate cancer (Goodrich et al, 2024; Lin et al, 2021; Wang et al, 2023); and food allergies and asthma (Hong et al, 2024).

Studies often use multiple omics technologies to comprehensively understand genetic and environmental effects of disease states (Choi et al, 2022; Maitre et al, 2022; Schmauch et al, 2023; Tebani et al, 2016; Zhang et al, 2024). Further, combining omics studies with longitudinal or cross-sectional studies allows for the ability to detect variability in individual exposures and identify critical windows of vulnerability (Bai et al, 2024).

### Sensors, geospatial devices, wearable, and mobile devices

Advances in other technologies such as sensors (e.g., measuring environmental pollutants and monitoring physiological parameters), geospatial devices (e.g., for monitoring locations), and

mobile devices (e.g., for measuring activity) enable exposure assessment. Exposure assessment quantifies the magnitude, frequency, and duration of exposure to environmental contaminants and indicates the population exposed. For the assessment of the external exposome, multiple tools can be employed. These include the geographical information system (GIS), satellite remote sensing, global positioning systems, and geolocation technologies, which can track a person's geographic position and potential exposures in different contexts (Beekhuizen et al, 2013; Seltenrich, 2014; Steinle et al, 2013; Turner et al, 2017). Portable and personal sensing, including smartphone-based sensors and assessments, is another significant avenue, providing real-time, high-resolution data about individual exposure to multiple environmental factors (Babu et al, 2024; Deville Cavellin et al, 2016; Loh et al, 2017; Nieuwenhuijsen et al, 2014; O'Connell et al, 2014; Snyder et al, 2013). Recent advancements in wearable exposome monitors have greatly facilitated personal exposome profiling and accelerated the progress of precision environmental health. Alongside these, self-reported questionnaire assessments are also used, with increasing reliance on internet-based platforms to capture self-reported, personal characteristics, and historic exposures (Dons et al, 2015; Dunton et al, 2014). Lastly, Geospatial modeling, which uses GIS techniques, predicts exposures based on spatial and temporal characteristics. It facilitates the creation of exposure maps to identify exposure hotspots and helps understand how exposure levels fluctuate over different regions and timeframes. These collective efforts significantly contribute to the comprehensive exposure assessment in epidemiology studies, aimed at gaining a better understanding of disease etiology. One application is the forecasting of allergenic pollen concentration for managing urban public health (Zhu et al, 2024).

A combination of environmental sampling, personal monitoring, and predictive mathematical models can help determine the amount of specific environmental agents a person or population is exposed to. This thorough analysis then establishes a connection between these environmental factors and molecular biomarkers, which facilitates comprehension of the interaction between molecular medicine and environmental exposures. In a pilot study, Gao et al integrated longitudinal personal exposome with internal multi-omics. They identified thousands of correlations between the exposome and the metabolome, proteome, and gut microbiome. In the participants studied, they found agrochemicals and fungi were predominant in the personal exposome, and the biomolecules and pathways related to the individual's immune system, kidney, and liver were highly associated with the personal external exposome (Gao et al, 2022). In a study by Herkert et al, wristbands were used to provide important exposure monitoring data. The study found that women had much higher levels of chemical exposures than men, and further analysis suggested that personal care products were the primary cause of these differences (Herkert et al, 2024).

These technologies are also helping underserved communities. Neighborhood sensors can reveal variations in exposomic factors such as air quality (Esie et al, 2022; Shatas and Hubbell, 2022), access to nutritious foods (An and Wang, 2023), and heat exposure (Li et al, 2024a; Lyu et al, 2022).

### Statistical and bioinformatic tools

The high-dimensional and complex data generated by exposomics require advanced statistical and bioinformatics tools for analysis and interpretation. Artificial intelligence (AI)-based approaches, such as machine learning (ML), neural networks, supervised and unsupervised learning, and deep learning, are increasingly being integrated into exposomics for pattern recognition, prediction modeling, data integration, risk evaluation, and pathway analysis (Shamji et al, 2023; Subramanian et al, 2020; Turner et al, 2017). The large and complex datasets from exposomics research are suitable for implementing these statistical and bioinformatic tools to predict disease risks such as asthma (Shamji et al, 2023), thunderstorm-triggered asthma (Kamangir et al, 2020), harmful algal blooms (Marrone et al, 2024), respiratory pandemics (Straub et al, 2024), and disaster recovery (Hanwacker, 2025).

## Exposome- and environment-wide association and gene-environment-wide interaction studies

Untargeted studies of the environment have been called exposome-wide association studies (ExWAS (Guillien et al, 2021) or XWAS (Feng et al, 2024)) or environment-wide association study (EWAS) (Patel et al, 2010). As the acronym EWAS is commonly used to denote epigenome-wide association studies, to avoid confusion, newer environment-wide association studies are using the acronym EnWAS (Sheehan et al, 2020). Those evaluating genetic changes associated with environmental exposures have been termed Gene × Environment (G × E), studies or gene-environment-wide interaction studies (GEWIS) (Shi et al, 2024).

In 2010, Patel et al conducted an environment-wide association study in which epidemiological data were comprehensively and systematically interpreted in a manner analogous to a Genome-Wide Association Study (GWAS). They performed multiple cross-sectional analyses associating 266 unique environmental factors with clinical status for type 2 diabetes. They found that the pesticide metabolite heptachlor epoxide, vitamin γ-tocopherol, and polychlorinated biphenyls were implicated in the onset of the disease (Patel et al, 2010). Another EnWAS study in the United States evaluated the annual use of 295 distinct pesticides and the incidence of prostate cancer and mortality rates in the United States. The study found a potential link between certain pesticides and increased prostate cancer incidence and mortality (Soerensen et al, 2024). A systematic review, however, found only 23 EnWAS articles published between January 2010 and December 2018. The authors cited limited exploitation of data sources, high heterogeneity in analytical approaches, and a lack of replication (Zheng et al, 2020).

An ExWAS study assessed 53 lifestyle/environmental factors in 599 adults and found that 21 joint lifestyle and environmental factors were associated with a low forced expiratory volume in one second (FEV1) in adults with asthma. When considered independently, none of the exposures revealed significant associations (Guillien et al, 2021). In another questionnaire-based ExWAS study, which used a questionnaire to assess exposures found that blood type A (Rh-) was associated with heart attack, biohazardous materials exposure with arrhythmia and higher paternal education level with reduced risk of multiple CVD outcomes (Lee et al, 2022).

A study integrated data from an exposome-wide association study and GWAS study on brain matter aging. They found that a number of environmental factors such as current tobacco smoking, length of mobile phone use, use of UV protection, and frequency of solarium/sunlamp were associated with brain matter aging. Several

**Table 1. Examples of major environmental/exposomic databases.**

| Name of database | Information on database | Organization | Link |
|---|---|---|---|
| Exposome-Explorer | First database dedicated to biomarkers of exposure to environmental risk factors for diseases | World Health Organization, International Agency for Research on Cancer. | http://exposome-explorer.iarc.fr |
| The Human Metabolome Database (HMDB) | Contains detailed information about small molecule metabolites found in the human body | Canadian Institutes of Health Research, Canada Foundation for Innovation, The Metabolomics Innovation Centre (TMIC), | https://hmdb.ca |
| METLIN | Lipids, steroids, plant and bacteria metabolites, small peptides, carbohydrates, exogenous drugs/metabolites, central carbon metabolites, and toxicants. | Scripps Research | https://metlin.scripps.edu/landing_page.php?pgcontent=mainPage: |
| Chemical Entities of Biological Interest (ChEBI) | 'small' chemical compounds | EMBL's European Bioinformatics Institute | https://www.ebi.ac.uk/chebi/ |
| PubChem | World's largest collection of freely accessible chemical information | National Library of Medicine | https://pubchem.ncbi.nlm.nih.gov |

single nucleotide polymorphisms (IP6K1, GMNC, OSTN, and SLC25A20) were significantly associated with the brain matter aging (Feng et al, 2024).

Other methodologies such as the gene-environment-wide interaction studies (GEWIS) aim to identify genetic loci with differential effects on the phenotype stratified by the levels of environmental exposure (Shi et al, 2024). These studies integrate the knowledge gained by environmental and genetic studies to further our understanding of the complex interplay of genetic and environmental factors in disease etiology These studies can be either hypothesis-driven using previous genetic and environmental knowledge, or agnostic for genetic information or agnostic for both genetic and environmental information. The paper by Kaufmann et al provides examples of GEWIS studies using the above strategies (Kauffmann and Demenais, 2012).

## Exposomic databases and networks

Exposomic research relies on a network of databases, centers, institutes, and other hubs to enable comprehensive data mining and analysis. Table 1 list some of the major exposomic databases.

### Centers, institutes, and hubs

The European Union continues to lead global efforts in exposomics. EXPoSOMICS was launched in 2012. It is a multidisciplinary project investigating associations between long-term exposure to pollutants and chronic diseases like cancer and cardiovascular conditions (Vineis et al, 2017). HELIX (The Human Early-Life Exposome) was launched in 2013 to examine early-life exposures and their effects on health from birth to adolescence. It used large existing birth cohorts and combines environmental data, biological samples, and omics analyses to comprehensively study the early-life exposome (Maitre et al, 2022). Helmholtz Munich Institute aims to integrate measures of internal and environmental exposures, lifestyle factors, and genetics to predict their combined influences on human health (Helmholtz Munich). Horizon Europe is the EU's key funding program for research and innovation and facilitates developing, supporting and implementing EU policies while tackling global challenges. It funded The European Human Exposome Network (EHEN), which was launched in 2020 to study the impact of environmental exposures on human health. It consists of nine large-scale projects (European Commission; Fayet et al, 2024). The International Human Exposome Network (IHEN) is another network which was built partly on the EHEN. Its aim is to build a worldwide network to bring together researchers, policymakers, and independent experts who can collaborate and improve human exposome research (The International Human Exposome Network, 2024).

In 2013, the HERCULES Exposome Research Center was launched by the National Institute of Environmental Health Sciences. It was the first exposome-based research center launched in the United States. Its vision is to serve as an intellectual hub in the advancement and translation of exposome research to improve human health. It is one of about 20 centers across the country dedicated to supporting all aspects of environmental health research at their home institutions and to developing collaborations with researchers across the country (Hercules Exposome Research Center, 2024). In 2015, an Exposome Workshop was held by the National Institute of Environmental Health Sciences to determine the current status of exposomics research and guide further inquiry. The workshop released a review of exposomics research covering the areas of (1) external exposure assessment, (2) biomonitoring, (3) biological response and impact, (4) epidemiology, and (5) data science. Three major recommendations for further advancement in the field of exposomics include infrastructure support, technology advancement, and promotion of exposomics. Infrastructure needs include an international exposome clearing house to promote data sharing, decrease redundancy, and share resources. The need to promote the sharing of databases, secondary analysis of banked samples and mining of databases. (Cui et al, 2016). This inspired the establishment of the Children's Health Exposure Analysis Resource (CHEAR), a database for laboratory and statistical studies on children and toxic environmental exposures (Balshaw et al, 2017). Children are at a higher risk for negative health outcomes as a result of toxic environmental exposures. This is primarily because they are still within critical developmental life stages that can easily be stunted or perturbed. CHEAR was key in the analysis of environmental factors and biological response indicators and provided support to over 30 studies measuring over 50,000 specimens, from 2015 to 2019. These

**Table 2. Major exposomic centers/projects.**

| | | |
|---|---|---|
| EXPoSOMICS | https://www.isglobal.org/en/-/exposomics | Europe |
| HELIX (The Human Early-Life Exposome) | https://www.projecthelix.eu/en/2013-02-06-15-56-29 | Europe |
| EHEN (European Human Exposome Network) | https://www.humanexposome.eu | Europe |
| The International Human Exposome Network (IHEN) | https://humanexposome.net | Europe |
| HERCULES | https://emoryhercules.com | United States |
| Human Health Exposure Analysis Resource (HHEAR) | https://hhearprogram.org/about-hhear | United States |
| Network for Exposomics in the U.S. (NEXUS) | https://www.nexus-exposomics.org | United States |

studies delved into the effects of toxic environmental exposures such as phthalates, phenols, metals, polyfluoroalkyl substances (PFAS), and flame retardants. They studied a multitude of adverse health outcomes, including asthma, diabetes, autism, obesity, and pregnancy complications (National Institute of Environmental Health Sciences). The Pediatric Research Using Integrated Sensor Monitoring Systems (PRISMS) was created in 2015 by the NIH to create sensor monitoring systems to measure the association between environmental, physiological, and behavioral factors and chronic diseases (National Institute of Biomedical Imaging and Bioengineering). In 2019, the Human Health Exposure Analysis Resource (HHEAR), created by the National Institute of Environmental Health Sciences, has been dedicated to the study of the exposome, including chemical, physical, biological, and social factors (National Institute of Environmental Health Sciences).

The Johns Hopkins Exposome Collaborative, established in 2019, has continued to be instrumental in operationalizing and implementing the objectives of exposomics (The Exposome Collaborative at Johns Hopkins University). Founded by Drs. Ramachandran and Hartung, it is involved in projects studying the association between internal and external exposures and asthma, specifically. They have also assessed sources of external toxic exposures, using personal and/or residential measurements of ultrafine particles, $PM_{2.5}$, metals, polycyclic aromatic hydrocarbons (PAHs), and other chemicals, along with internal exposures, which are quantified using plasma, serum, and urine samples. In 2024, the National Institutes of Health (NIH) created a $7.7 million center, the Network for Exposomics in the United States (NEXUS) to build a global community of practice for cooperation and collaboration in coordination with the International Human Exposome Network (Environmental Factor). Table 2 list some of the major exposomic centers and projects.

## Challenges and future directions

The lack of standardization of exposome cohorts, exposure assessment, and outcome assessment is one of the critical challenges in exposomics research (Haddad et al, 2019). Other critical challenges that have been identified include the replication and validation of findings, the creation of statistical and computational methods, and the practical translation of exposome research to clinical practice and disease prevention (Siroux et al, 2016; Turner et al, 2017; Zhang et al, 2021).

Currently, there is little published on the ethical, legal, and social issues in the field of exposomics. A study identified five ethical themes pertaining to exposomics: the goals of exposome research, its standards, its tools, how it relates to study participants, and the consequences of its products. They also highlighted three aspects of exposome research most in need of ethical reflection: the actionability of its findings, the epidemiological or clinical norms applicable to exposome research, and the meaning and action–implications of bias (Safarlou et al, 2023). Existing literature has identified ethical and regulatory factors that shape genomics research, personal health information, and risks for biomedical research (Safarlou et al, 2023). However, the issues that arise from exposomics research are complex and intertwined with law, policy, and regulations. These issues include the exacerbation of inequities if precision health is only focused on high-resource settings (Hekler et al, 2020), increasing inequality through the deviation of those who can afford personalized medicine (Cesario et al, 2021), concerns regarding individual genetic discrimination (Rabinowitz and Poljak, 2003), changes to the application of the environmental tort law (Dyke et al, 2019), and the expansion of human rights and ethics declarations to protect against disparities (Dupras et al, 2020).

To meet these challenges, committed investments for exposomics research and massive consortium building should occur on a huge and global scale. Biobanks, data storage, statistical, bioinformatic tools, and computational tools need to be shared. Regulations to safeguard privacy should be put in place (Grady et al, 2023).

## Conclusion

The exposome offers a holistic lens to view the complex interplay between environmental exposures, genetic factors, and molecular pathways in disease etiology and progression. By considering the totality of exposures across the lifespan, exposomics complements genomics and provides a more comprehensive understanding of diseases and their multifactorial nature. (Barouki et al, 2018) Exposomics uses the tools and information from different fields such as environmental science, epidemiology, computational science, artificial intelligence, and molecular medicine to understand in totality how the exposome affects human health (Fig. 2). In the future, by combining information from both the genome and exposome, one can better assess an individual's risk of diseases and personalize frequency of monitoring, and preventative and treatment plans. It can also assist communities, and policymakers to implement measures that mitigate harmful exposures and

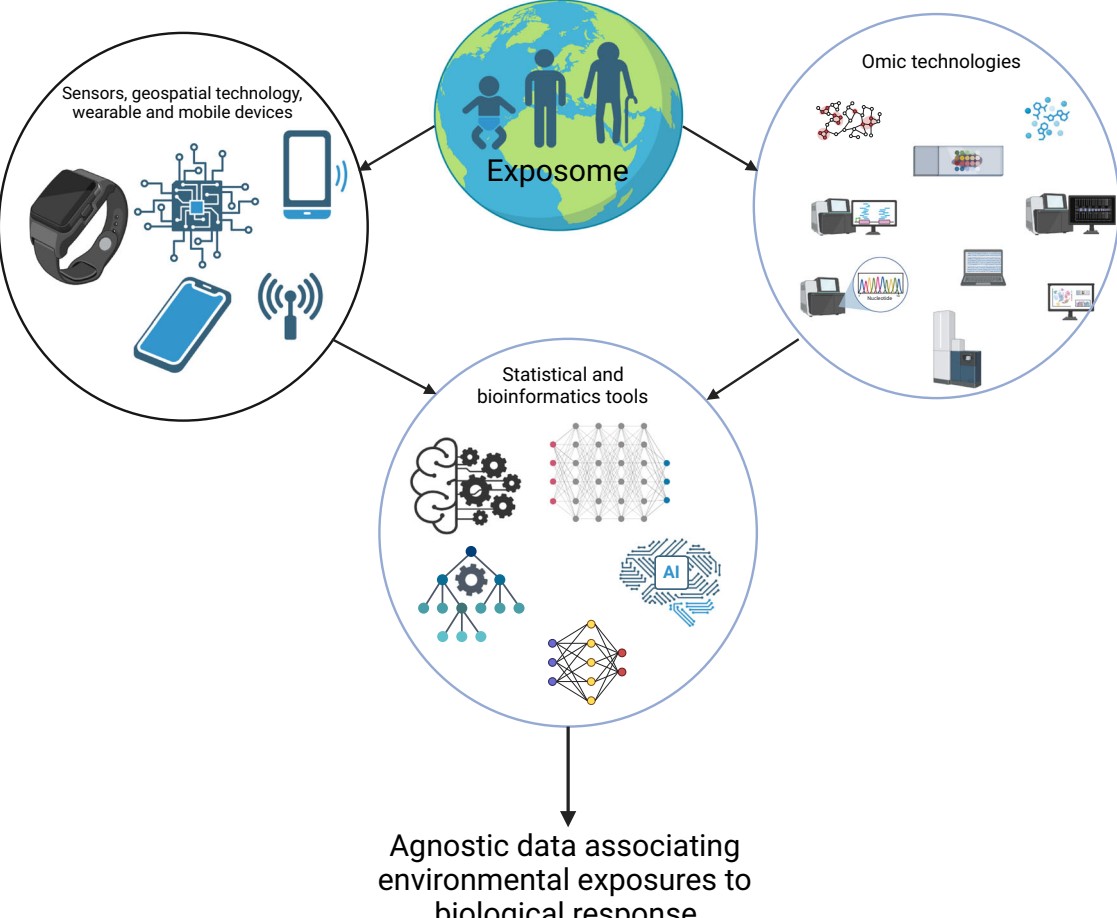

**Figure 2. Data integration in exposomics.**

Exposomics is dependent on data from a number of high-throughput technologies as well as information from sensors, geospatial devices, and wearable and mobile devices. The data were then processed and analyzed by statistical and bioinformatic tools that can provide associations between exposures and biological responses.

promote healthier environments. However, the field is still in its infancy. There are many challenges ahead.

## Pending issues

- Standardization of exposome cohorts, exposure assessment, and outcome assessment.
- Ethical, legal, and social issues related to exposomics data.
- Improvements in statistical and computational methods.
- Practical translation of exposomic research into clinical practice and disease prevention.
- Financing data storage capabilities.
- Harmonizing and sharing data.

## Peer review information

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

## Acknowledgements

NIAID grants R21AI1492771, R21EB030643, U01AI140498, U01 AI147462, R01AI140134, UM1AI109565, UM2AI130836, P01AI153559, U19AI167903, R01 AI125567, R21AI149277, NHLBI grants P01 HL152953 and R01 HL141851, NIEHS grants R21ES03304901 and R01 ES032253.

## Author contributions

Melissa Wan: Writing—original draft; Writing—review and editing. Elisabeth M Simonin: Writing—original draft; Writing—review and editing. Mary Margaret Johnson: Writing—original draft; Writing—review and editing. Xinyue Zhang: Writing—original draft; Writing—review and editing. Xiangping Lin: Writing—original draft; Writing—review and editing. Peng Gao: Writing—original draft; Writing—review and editing. Chirag J Patel: Writing—original draft; Writing—review and editing. Aroub Yousuf: Writing—original draft; Writing—review and editing. Michael P Snyder: Writing—original draft; Writing—review and editing. Xiumei Hong: Writing—original draft; Writing—review and editing. Xiaobin Wang: Writing—original draft; Writing—review and editing. Vanitha Sampath: Visualization; Writing—original draft; Writing—review and editing. Kari C Nadeau: Conceptualization; Resources; Supervision; Funding acquisition; Writing—original draft; Project administration; Writing—review and editing.

## Disclosure and competing interests statement

XZ is a cofounder of Exposomics, Inc. MPS is a cofounder and scientific advisor of Crosshair Therapeutics, Exposomics, Filtricine, Fodsel, iollo, InVu Health, January AI, Marble Therapeutics, Mirvie, Next Thought AI, Orange Street Ventures, Personalis, Protos Biologics, Qbio, RTHM, SensOmics. MPS is a scientific advisor of Abbratech, Applied Cognition, Enovone, Jupiter Therapeutics, M3 Helium, Mitrix, Neuvivo, Onza, Sigil Biosciences, TranscribeGlass, WndrHLTH, Yuvan Research. MPS is a cofounder of NiMo Therapeutics. MPS is an investor and scientific advisor of R42 and Swaza. MPS is an investor in Repair Biotechnologies. All other authors declare no conflict of interest.

