## [Peer Review File · EMBO Molecular Medicine]

Exposomics: A Review of Methodologies, Applications, and Future Directions in Molecular Medicine

Melissa Wan, Elisabeth Simonin, Mary Johnson, Xinyue Zhang, Xiangping Lin, Peng Gao, Chirag Patel, Aroub Yousuf, Michael Snyder, Xiumei Hong, Xiaobin Wang, Vanitha Sampath, and Kari Nadeau

Corresponding author(s): Kari Nadeau (knadeau@hsph.harvard.edu)

Review Timeline:

Submission Date:	23rd Jun 24
Editorial Decision:	9th Sep 24
Revision Received:	6th Dec 24
Accepted:	24th Dec 24

Editor: Zeljko Durdevic

Transaction Report:

9th Sep 2024

Dear Dr. Nadeau,

Thank you for the submission of your manuscript to EMBO Molecular Medicine, and please accept my apologies for the unusual delay in getting back to you. We have now received feedback from the two of the three reviewers who agreed to evaluate your manuscript. We were hoping to receive comments from the referee #2, however after several vain promises from the referee we decided to make a decision now in order to avoid further delay in the process.

As you will see from the reports below, the referees are positive about its interest and timeliness, however, they also raise serious criticisms that should be addressed in a revised manuscript. Further consideration of a revision that addresses reviewers' concerns in full will entail a second round of review. Particular focus should be given to revising the manuscript to provide more systematic and balanced view on the achievements and limitations of exposomic studies as suggested by the referee #1.

I would also like to ask you to add the following items to your revised article:

- 1) Change "molecular biology" to "molecular medicine" in the title and in the Figure 1 legend.
- 1) Add "Disclosure and competing interests statement". We updated our journal's competing interests policy in January 2022 and request authors to consider both actual and perceived competing interests. Please review the policy <https://www.embopress.org/competing-interests> and update your competing interests if necessary.
- 2) Glossary: The glossary is meant to explain some of the terms used for laymen. Could you please identify terms that may need an "explanation"?
- 3) Pending issues: At the end of each article is a box highlighting issues that still need further studies and where research efforts should converge. Could you identify some pending issues?
- 4) Funding: Please make sure that information about all sources of funding are complete in both our submission system and in the manuscript in "Acknowledgments".
- 5) As part of the EMBO Publications transparent editorial process initiative EMBO Molecular Medicine will publish online a Review Process File (RPF) to accompany accepted manuscripts. This file will be published in conjunction with your paper and will include the anonymous referee reports, your point-by-point response and all pertinent correspondence relating to the manuscript. Let us know whether you agree with the publication of the RPF.

You can submit your revised files by logging onto our online manuscript tracking system or simply follow this link:

Link Unavailable

I hope that the referees' comments do not prove too problematic to address and I look forward to reading your next version.

Yours sincerely,

Zeljko Durdevic

*** IMPORTANT INFORMATION ***

- 1) a .doc formatted version of the manuscript text (including Figure legends and tables)
- 2) Separate figure files
- 3) a letter INCLUDING the reviewer's reports and your detailed responses to their comments.

Also, and to save some time should your paper be accepted, please read below for additional information regarding some features of our research articles:

1) Glossary: EMBO Molecular Medicine articles will be accompanied by a glossary explaining some of the terms used for laymen. I identified the following:

_____, _____, _____

Could you please help us in identifying terms that may need an "explanation" other terms that we can add to the glossary.

2) Pending issues: At the end of each article we will have a box highlighting issues that still need further studies and where research efforts should converge (we call this the Pending issues box). From my reading I would say:

but I can see there may be many more. Could you work on this as well?

3) Disclosure and competing interest statement: Please include a statement declaring any competing commercial interests in relation to your submitted work.

4) Please note that we now mandate that all corresponding authors list an ORCID digital identifier. This takes <90 seconds to complete. We encourage all authors to supply an ORCID identifier, which will be linked to their name for unambiguous name identification.

Currently, our records indicate that the ORCID for your account is 0000-0002-2146-2955.

Link Not Available

-

Thank you,

Zeljko Durdevic

***** Reviewer's comments *****

Referee #1 (Remarks for Author):

This is a narrative review of the applications of exposomics with several limitations:

1. It is entirely centred on the United States, ignoring the large grants awarded by the European Commission and coordinated through the EHEN and IHEN networks
2. The description of bottom-up and top-down approaches is scholastic and does not reflect the nuances of the original papers
3. Most of the text is very optimistic/acritical towards the achievements of the exposome approach, and there is no attempt to make a critical appraisal of what has been fulfilled and what not
4. Much of the last part is based on a single review (Haddad Andrianou & Makris, 2019), with some questionable statements (e.g. many studies have looked at metabolites and psychological stress, contrary to what they say)
5. More generally, this is not a systematic review and is acritical and superficial in its account of the contributions of exposome science, for example:

" Finally, the microbiome is also a key player in the communication between external exposures and metabolites and reflects the body's response to external stressors, such as diet, air pollution and heavy metals"

Or: "Applications of exposomics research can enable the creation of targeted interventions, promote healthy environments, mitigate harmful exposures, and reduce health disparities through creation of programs and policy changes".

Referee #3 (Remarks for Author):

This review written by Wan et al. entitled "Exposomics: A Review of Methodologies, Applications, and Future Directions in molecular Biology" summarized the current exposomic-related studies, provided an overall picture of the updated knowledge in this field and proposed the possible future directions. For content integrity and readability, several suggestions are listed below:

1. In the section of "Foundation of Exposomics", if the authors could provide a more detailed explanation of the "bottom-up" and "top-down" approaches for characterizing the total exposome, the readers may better understand the difference between these two methods as mentioned by the authors.
2. In the line of 98, the novel approach reported by Patel is discussed. The authors described that this approach has been increasingly adopted, especially in prenatal and pediatric population. However, it is unclear what approach was used to estimate exposome. Author should make contents more concrete, not only just the multi-factorial lens.
3. In line of 106, "three major recommendations" is mentioned. How or what aids are available for further advancements in the field of exposomics? For example, what is the infrastructure support, and how is it used to advanced exposomics?
4. In the section of "Methodological approaches", the authors describe the use of omics technology in exposomics. Omics include various molecular levels such as epigenomics, genomics, transcriptomics, proteomics, and metabolomics. However, the authors only describe GC-MS, LC-MS and similar technologies, which are used to profile proteomics or metabolomics. The authors seem to focus solely on metabolism. They should add more content to describe the technologies or methods used in omics, how did they aid in exposomics studies, and which studies have used these methodologies?
5. What is the meaning of biomonitoring? Is it the equipment to monitor the environment exposure? Based on the content, it seems to refer to experimental equipment used to detect chemicals or metabolites in specimen (blood or urine) using traditional (targeted) or novel (untargeted) methods. How is "biomonitoring" different from the "omics technology"? The authors should clarify that what technology is to measure the chemical compounds in the specimens, while "omics technology" measures different molecular levels.
6. The authors describe geospatial modeling as useful for exposure assessment. Are there any studies that have applied this methodology to study environmental exposures and human health?
7. It would be highly beneficial to draw a figure that integrates the key concepts, important factors, and necessary tools for exposomics studies. This figure would include elements such as cohorts, exposure assessment, platforms, technologies, computational tools, data, and algorithms as described by the authors in the article. This visual representation would help guide further research and provide a clearer understanding the interconnected components in exposomics studies.

Response to Reviewer's comments

Reviewer 1

Comment 1: This is a narrative review of the applications of exposomics with several limitations: It is entirely centred on the United States, ignoring the large grants awarded by the European Commission and coordinated through the EHEN and IHEN networks

- **Author Response:** *Thank you for your time and review of the manuscript. We have added information on European exposomics projects including EHEN and IHEN networks. (Page 9, Lines 310-327)*

Comment 2: The description of bottom-up and top-down approaches is scholastic and does not reflect the nuances of the original papers

- **Author Response:** *We have revised and clarified the bottom-up and top-down approaches.*
 - *Rappaport and Smith in 2010 proposed another way to approach exposomics research. They contrasted two approaches to characterizing the total exposome, the top-down and the bottom-up approaches. The top-down approach measures important exposure-related biomarkers within biospecimens that are biologically relevant and mediate health effects. It does not capture direct measures of exposure and can generate hypotheses regarding exposure and biological responses. It uses untargeted omics methods to measure both exogenous and endogenous exposures. The bottom-up approach comprehensively measures environmental exposures including chemical, physical, and social factors. This data can be obtained through various sources such as geospatial monitoring, surveys, census and other public databases, wearable and portable devices, and social media platforms. While this provides valuable environmental data, it fails to address the internal chemical environment. Both the bottom-up and top-down are complementary and each approach provides valuable data. (Page 5, Lines 145-157)*

Comment 3: Most of the text is very optimistic/acritical towards the achievements of the exposome approach, and there is no attempt to make a critical appraisal of what has been fulfilled and what not

- **Author Response:** *We agree and we have a section on challenges where we have discussed the limitations and challenges of exposomics. (Pages 10-11, Lines 374-398)*

Comment 4: Much of the last part is based on a single review (Haddad Andrianou & Makris, 2019), with some questionable statements (e.g. many studies have looked at metabolites and psychological stress, contrary to what they say)

- **Author Response:** *We have removed this section.*

Comment 5: More generally, this is not a systematic review and is acritical and superficial in its account of the contributions of exposome science, for example: " Finally, the microbiome is also a key player in the communication between external exposures and metabolites and reflects the

body's response to external stressors, such as diet, air pollution and heavy metals" Or: "Applications of exposomics research can enable the creation of targeted interventions, promote healthy environments, mitigate harmful exposures, and reduce health disparities through creation of programs and policy changes".

- **Author Response:** *We have extensively revised the manuscript and provided a more in-depth review of exposomics. We have provided examples of studies on exposomics throughout to substantiate the statements made. Here are a few examples:*
 - *In another study, urine samples were collected from sugarcane workers to determine environmental exposures leading to the high rate of kidney disease seen in these workers. Samples underwent apply exposomic and untargeted metabolomic analysis. The study results suggest that silica and certain pesticides were significantly elevated in the urine of sugarcane workers. These exposures may provide insight into early warning signs of kidney injury and may help explain the increased incidence of chronic kidney disease among these workers. (Page 6, Lines 195-201)*
 - *In a study by Herkert et al., wristbands were used to provide important exposure monitoring data. The study found that women had much more higher levels of chemical exposures than men and further analysis suggested that personal care products were the primary cause of these differences (Page 7, Lines 244-247)*

Reviewer 3

Comment 1: This review written by Wan et al. entitled "Exposomics: A Review of Methodologies, Applications, and Future Directions in molecular Biology" summarized the current exposomic-related studies, provided an overall picture of the updated knowledge in this field and proposed the possible future directions. For content integrity and readability, several suggestions are listed below:

- **Author Response:** *Thank you for your time and comments regarding the manuscript.*

Comment 2: In the section of "Foudation of Exposomics", if the authors could provide a more detailed explanation of the "bottom-up" and "top-down" approaches for characterizing the total exposome, the readers may better understand the difference between these two methods as mentioned by the authors.

- **Author Response:** *We have revised and clarified the bottom-up and top-down approaches. (Page 5, Lines 145-157)*

Comment 3: In the line of 98, the novel approach reported by Patel is discussed. The authors described that this approach has been increasingly adopted, especially in prenatal and pediatric population. However, It is unclear what approach was used to estimate exposome. Author should make contents more concrete, not only just the multi-factoral lens.

- **Author Response:** *We have revised this section and added more details (Page 8, lines 272-277)*
 - *In 2010, Patel et al. conducted an EWAS study in which epidemiological data were comprehensively and systematically interpreted in a manner analogous to a*

Genome Wide Association Study (GWAS). They performed a multiple cross-sectional analyses associating 266 unique environmental factors with clinical status for type 2 diabetes. They found that the pesticide metabolite heptachlor epoxide, vitamin γ -tocopherol, and polychlorinated biphenyls were implicated in the onset of the disease

Comment 4: In line of 106, "three major recommendations" is mentioned. How or what aids are available for further advancements in the field of exposomics? For example, what is the infrastructure support, and how is it used to advanced exposomics?

- **Author Response:** *We have elaborated on the recommendations*
 - *Infrastructure needs include an international exposome clearing house to promote data sharing, decrease redundancy, and share resources. The need to promote sharing of databases, secondary analysis of banked samples and mining of databases. (Page 10, Lines 336-342)*

Comment 5: In the section of "Methodological approaches", the authors describe the use of omics technology in exposomics. Omics include various molecular levels such as epigenomics, genomics, transcriptomics, proteomics, and metabolomics. However, the authors only describe GC-MS, LC-MS and similar technologies, which are used to profile proteomics or metabolomics. The authors seem to focus solely on metabolism. They should add more content to describe the technologies or methods used in omics, how did they aid in exposomics studies, and which studies have used these methodologies?

- **Author Response:** *We have revised the manuscript to include this information.*
 - *The rise of "omics" technologies such as genomics, metabolomics, DNA adductomics, proteomics, lipidomics, transcriptomics, microbiomics, and epigenomics offer comprehensive and detailed insights into the genetic, epigenetic, molecular and cellular responses to environmental exposures. These technologies have been made possible by the availability and advancement of high-throughput analytical instruments, such as DNA, RNA and protein microarrays, Nuclear Magnetic Resonance (NMR) Spectroscopy, next generation sequencing (NGS), Bisulfite Sequencing, ATAC-Seq, Single-cell RNA Sequencing, Chromatin Immunoprecipitation Sequencing (ChIP-Seq), and mass spectrometry (MS). Moreover, when combined with additional separation technologies like gas chromatography-mass spectrometry (GC-MS) and liquid chromatography-mass spectrometry (LC-MS), the utility of MS is further enhanced, enabling the detection, quantification, and identification of a broad spectrum of environmental exposures, and the associated biological responses, with increased sensitivity. (Page 5, Lines 164-176)*
- **Comment 6:** What is the meaning of biomonitoring? Is it the equipment to monitor the environment exposure? Based on the content, it seems to refer to experimental equipment used to detect chemicals or metabolites in specimen (blood or urine) using traditional (targeted) or novel(untargeted) methods. How is "biomonitoring" different from the

"omics technology"? The authors should clarify that what technology is to measure the chemical compounds in the specimens, while "omics technology" measures different molecular levels.

- **Author Response:** *We have removed the use of the term biomonitoring.*

Comment 7: The authors describe geospatial modeling as useful for exposure assessment. Are there any studies that have applied this methodology to study environmental exposures and human health?

- **Author Response:** *Yes. One application is the forecasting of allergenic pollen concentration for managing urban public health. (Page 7, Lines 231-233)*

Comment 8: It would be highly beneficial to draw a figure that integrates the key concepts, important factors, and necessary tools for exposomics studies. This figure would include elements such as cohorts, exposure assessment, platforms, technologies, computational tools, data, and algorithms as described by the authors in the article. This visual representation would help guide further research and provide a clearer understanding the interconnected components in exposomics studies.

- **Author Response:** *We have added the following figure.*

Sincerely,

Kari C Nadeau

Kari Nadeau, MD, PhD
Department Chair, Dept of Environmental Health (link to website)
John Rock Professor of Climate and Population Studies
Director of the Allergy, Climate, and Exposomics Lab at Harvard (link to website)

24th Dec 2024

Dear Prof. Nadeau,

Please find enclosed the final reports on your manuscript. We are pleased to inform you that your manuscript is accepted for publication and is now being sent to our publisher to be included in the next available issue of EMBO Molecular Medicine.

Your manuscript will be processed for publication by EMBO Press. It will be copy edited and you will receive page proofs prior to publication.

There is no charge for this Review Article. However, in a few weeks, when you are contacted to sign your license agreement and review the article proofs, you will need to enter a token into the appropriate field in the Springer Nature Author Services system. Please note that we will provide the token in a separate letter. Be aware that, due to the holiday season, we anticipate a delay in processing your manuscript.

Referee #3 (Remarks for Author):

The authors have adequately addressed the critiques.

Referee #3 (Remarks for Author):

The authors have adequately addressed the critiques.